# Chromosome splitting of *Plasmodium berghei* using the CRISPR/Cas9 system

**Daniel Addo-Gyan**[1]☯, **Haruka Matsushita**[1]☯, **Enya Sora**[1], **Tsubasa Nishi**[2], **Masao Yuda**[2], **Naoaki Shinzawa**[1], **Shiroh Iwanaga**[3,4]*

**1** Department of Environmental Parasitology, Graduate School of Medical and Dental Sciences, Tokyo Medical and Dental University, Bunkyo-ku, Tokyo, Japan, **2** Laboratory of Medical Zoology, Department of Medicine, Mie University, Kurimamachi Yacho, Tsu, Mie, Japan, **3** Department of Molecular Protozoology, Research Institute for Microbial Diseases, Osaka University, Yamadaoka, Suita, Osaka, Japan, **4** Center for Infectious Disease Education and Research (CIDER), Osaka University, Yamadaoka, Suita, Osaka, Japan

☯ These authors contributed equally to this work.
* iwanaga@biken.osaka-u.ac.jp

**Data Availability Statement:** All relevant data are within the paper and its Supporting Information files.

## Abstract

*Spatial arrangement of chromosomes is responsible for gene expression in Plasmodium* parasites. However, methods for rearranging chromosomes have not been established, which makes it difficult to investigate its role in detail. Here, we report a method for splitting chromosome in rodent malaria parasite by CRISPR/Cas9 system using fragments in which a telomere and a centromere were incorporated. The resultant split chromosomes segregated accurately into daughter parasites by the centromere. In addition, elongation of *de novo* telomeres were observed, indicating its proper function. Furthermore, chromosome splitting had no effect on development of parasites. Splitting of the chromosome is expected to alter its spatial arrangement, and our method will thus be useful for investigating its biological role related with gene expression.

## Introduction

*Plasmodium* parasites, which are causative agents of malaria, possess a complex life cycle consisting of distinctive developmental stages between mosquitos and animals. Each developmental stage is controlled by the stage-specific gene regulation, for which sequence-specific transcription factors [1] and epigenetic regulators are responsible [2, 3]. In addition to these, the spatial arrangement of chromosomes in the nucleus is recently considered to participate in the regulation of gene expression in parasites. For instance, chromosome conformation capture analysis using next-generation sequencing (Hi-C) has shown that heterochromatic regions scattered throughout chromosomes form a cluster at the periphery of the nucleus, which are known as a repressive center [4, 5]. Dissociation from the repressive center will change the chromatin state from heterochromatin to euchromatin, which triggers activation of transcription [6, 7]. Since multi-gene families of infected red blood cell (RBC)-surface antigens and the sex-specific transcription factor are located in these heterochromatic regions [8], their dissociation may be responsible for antigenic variation and sexual development.

**Funding:** All studies were supported by Grants-in-Aid for Scientific Research (20H03477 and 19K22527 to S.I. 18K07084 and 21K06985 to N.S., and 17H01542 to M.Y.), which were funded by the Japan Society for the Promotion of Science (JSPS: https://www.jsps.go.jp/), and also supported by the Japan Agency for Medical Research and Development (AMED: https://www.amed.go.jp/) under Grant Number 21jm0210061h0004 (to S.I.), 21wm0325018 (to N.S.) and 21wm0225014 (to N. S.). The funders have no role in study design, data collection and analysis, decision to publish, or preparation of the manuscript.

**Competing interests:** The authors have declared that no competing interests exist.

Furthermore, the three-dimensional size and volume of chromosomes change during the progression of asexual development in RBC, which may be responsible for the changes in transcriptional activity [4]. To investigate the biological role of the spatial arrangement of chromosomes in gene expression, engineering large genomic regions is required. However, there are no established methods in *Plasmodium* parasites at present.

The CRISPR/Cas9 system is a useful technique for engineering the genes of *Plasmodium* parasites [9]. In this system, a gene is modified through two steps as follows: the targeted genomic locus cleavage by the Cas9-single guide RNA (sgRNA) complex, and the induced double-strand break repair by homology-directed recombination (HDR) using donor template DNA. In our previous study, we generated the Cas9-expressing rodent malaria parasite (*P. berghei*), that successfully engineered the genes with high efficiency, by transfecting these parasites with linear donor template DNA, and the plasmid encoding sgRNA [10]. With this system, we were able to remove more than 50 kbp of the subtelomere region of chromosome 1 [10]. Briefly, we cleaved the border of the subtelomere and non-telomeric chromosomal region on chromosome 1, followed by HDR providing the telomeric sequence at the cleaved end of the chromosome. The newly generated telomere functioned properly, which ensured the replication of the *de novo* chromosome end. The resultant transgenic parasites that carries the truncated chromosome 1 had no growth defects during both asexual and sexual development in RBC. This truncation of the chromosome suggests that the large-scale editing of chromosomes could be achieved by the CRISPR/Cas9 system.

In this study, we developed a method for splitting the chromosome using the CRISPR/Cas9 system. In addition to a telomere, we used a centromere which is responsible for chromosome segregation into daughter cells during nuclear division. After confirming the split of the chromosome, its effect on the development of the parasites was examined. Furthermore, to test the versatility of this method, we attempted to cleave the chromosome at different loci. Our method enables us to split chromosomes in a flexible manner, which allows for a variety of future applications including large-scale genome editing.

## Materials and methods

### Animal experiments

All animal experiments were carried-out in accordance with the guidelines for the care and use of laboratory animals, approved by the animal experimentation committee of the Tokyo Medical and Dental University. For all experiments requiring the use of total mouse blood, mice were deeply anesthetized using isoflurane, followed by collection of total blood by cardiac puncture, and euthanized by cervical dislocation immediately after collection. At the end of all mice experiments, euthenization was performed using 70% carbon dioxide and 30% oxygen mixture in a gas chamber, ensuring that mice were completely dead before removal from the chamber.

### Construction of plasmid having the guide RNAs

A 19-bp sequence of guide RNA (gRNA) was designed upstream of the protospacer-adjacent motif (PAM), and a pair of complementary oligonucleotides was synthesized for each target site. The gRNA was transcribed by PfU6 (U6 spliceosomal RNA, PF3D7_1341100) promoter. Since the PfU6 promoter requires a guanosine nucleotide to initiate transcription, a guanosine was added at the 5' end of the designed oligonucleotide that encoded the sense sequence. In addition, the oligonucleotides were designed to generate overhangs to be used for cloning into *Bsm*BI-digested psgRNA1 plasmid [10]. The cloned gRNA was placed under PfU6 promoter and fused with a tracrRNA, which generated an sgRNA. The psgRNA1 plasmid has human

dihydroreductase gene as drug selectable marker. Thus, the transfected parasite with this plasmid can be selected using pyrimethamine. The designed oligonucleotides were shown in S1 Table.

### Preparation of linear donor templates with telomere and centromere

The DNA fragment used for HDR was amplified by PCR using primer set (S1 Table) and then cloned upstream of telomere sequence of pArm_L plasmid (S1 Fig). The cloned DNA fragment was excised together with the telomere sequence by restriction digestion of 10 μg of resultant plasmid with *Sal*I and *Pme*I. This linear DNA fragment was used for HDR with the centric chromosome (Chromosome 1 fragment with the original centromere after cleavage) after cleavage by the Cas9-sgRNA complex. Another DNA fragment, which was used for HDR with the acentric chromosome (Chromosome 1 fragment lacking a centromere after cleavage), was amplified and cloned upstream of both telomere and centromere of pArm_R plasmid (S1 Fig). The linear DNA fragment with the centromere and telomere was excised form the resultant plasmid by digestion with *Sal*I and *Pme*I, and used for the transfection experiment.

### Transfection of parasites

Transfection of parasites had been described previously [11]. Briefly, two linear forms of DNA fragments that had a telomere only, and both telomere and centromere were co-introduced with the psgRNA1 plasmid having the sgRNA, into purified schizonts ($1 \times 10^7$ parasites) of *P. berghei* ANKA using the parasite nucleofector II kit and the Nucleofector II device with the U-033 program (Lonza). Transfected parasites were injected intravenously into 5–7 weeks old female ddY mice, immediately after electroporation. Treatment with pyrimethamine was initiated 30 hours post- infection and continued for 5 days, followed by withdrawal of the drug. Transfection experiments for generating transgenic parasites were carried out independently in duplicate. The clonal transgenic parasite lines were obtained by limiting dilution procedure.

### Contour-Clamped Homogenous Electric Field (CHEF) electrophoresis analysis

Whole blood from mice inoculated with transgenic parasites were used to make DNA agarose plugs (at approximately $1.0 \times 10^8$ parasites / plug) as described in a previous study [12]. Chromosomes were separated on a 1% pulse-field certified agarose gel on the CHEF Mapper XA system (Bio-Rad) under the following conditions: 0.5 x TBE buffer, temperature: 14˚C, switch time: 10–80 sec, runtime: 20 hours, included angle: 120˚, voltage gradient: 6 V/cm.

### Southern hybridization analysis

Whole blood from mice inoculated with transgenic parasites was filtered through the cellulose powder D columns (Advantec, #49020040) to remove leucocytes. The RBCs were collected by centrifugation and lysed with a lysis buffer (1.5 M $NH_4Cl$, 0.1 M $KHCO_3$, and 0.01 M EDTA) to obtain the parasites. The genomic DNA was purified from the obtained parasites as described previously [12]. The genomic DNA purified from split-Ch1-1053 and split-Ch1-1116 parasites were digested with *Eco*RI, followed by blotting onto a nylon membrane. The probe DNA was labelled with DIG according to the manufacturer's instructions (Roche Diagnostics GmbH) and used for hybridization with genomic DNA blotted on the membrane. Signals derived from hybridized DNA was detected using the Chemidoc MP system (Bio-rad).

Chromosomes separated by CHEF electrophoresis were transferred onto a nylon membrane and hybridized with probes specific for PBANKA0112500 and PBANKA_0104900. Subsequent Southern hybridizations were performed in a similar manner as described above.

### Evaluation of asexual multiplication and merozoite formation

Asexual multiplication in RBCs was evaluated by monitoring parasitemia in infected mice. The 1,000 iRBCs were injected intravenously into naive mice and the progress of the parasitemia was examined every 12 hours using a Giemsa-stained thin blood smear. All experiments were performed in triplicates. Averages of parasitemia between split-Ch1-1116 and parental pbcas9 parasites were compared using a *t*-test. The growth rate was calculated based on the approximate growth curve. The curve was represented by the following equation;

$$P = Ae^{xD} \tag{1}$$

Where P is the parasitemia; A is the constant value; D is the day post-infection; and $e^x$ is the growth rate.

The mice were inoculated with split-Ch1-1116 intra-peritoneally. At a parasitemia of 2–3%, whole blood taken by cardiac puncture was washed once with 20 ml RPMI medium containing 20% fetal calf serum (FCS). The infected RBCs were cultured with 25 ml of culture medium (RPMI 1640, 20% FCS, 500 μL penicillin-streptomycin) under low oxygen conditions: 5% oxygen, 5% carbon dioxide, 90% nitrogen at 37˚C for 16 hours. Giemsa-stained thin smears were prepared and the number of merozoites per mature schizont was counted for 100 schizonts.

### Evaluation of exflagellation and ookinete formation

Exflagellation of male gametes was assessed by counting the number of exflagellation centres. Mice were pre-treated with an intraperitoneal injection of 0.2 ml phenylhydrazine (6 mg/ml in PBS) 3 days prior to parasite infection to stimulate reticulocyte formation. 1 x 10^5 iRBCs were intravenously inoculated into the phenylhydrazine-treated mice. At 5 days post-infection, infected blood was collected from the tail vein and diluted twenty-fold with ookinete culture medium (RPMI1640 containing 100 μM xanthurenic acid, 50 mg/l hypoxanthine, 25 mM HEPES, 24 mM $NaHCO_3$, 50 U/ml penicillin, 50 μg/ml streptomycin, and 20% FCS, adjusted to pH 7.5). The number of exflagellation centres per 10,000 RBC was counted using Burker-Turk counting chambers after incubation at 20˚C for 10 min.

Ookinete formation was evaluated as previously described [13]. Briefly, mice were pre-treated with phenylhydrazine and infected as in the exflagellation assay described above. At 5 days post-infection, infected blood was collected from mice, and leukocytes were removed using cellulose powder D columns. Leukapheresis blood was diluted 10-fold with ookinete culture medium. Diluted blood samples were incubated at 20˚C for 22–24 h. Giemsa-stained thin smears were prepared and assessed for ookinetes. The number of mature and immature ookinetes were counted, and the mature ookinete rate was calculated as follows:

$$Mature\ ookinete\ rate = \left( \frac{number\ of\ mature\ ookinetes}{Number\ of\ mature\ ookinetes + number\ of\ immature\ ookinetes} \right) x100 \tag{2}$$

## Results

### Splitting chromosome 1 of *Plasmodium berghei*

In this study, we selected chromosome 1 as a target for proving the experimental concept of chromosome splitting. The chromosome 1 of *P. berghei* consists of 515,659bp and 136 genes. The centromere of this chromosome is located at 389,413–390,757, and the sub-telomeres are located within less than 50 kbp from both ends (https://plasmodb.org/). To split the chromosome, we utilized the CRISPR/Cas9 system using the donor template DNAs in which telomere and a centromere sequences were incorporated. The telomeres of *P. berghei* are composed of a repetition of a degenerated motif (5'-GGGTTYA, where Y is T or C) [14], and the centromeres

of this parasite are highly A/T rich sequences and are the smallest regional centromeres, of which sizes range from 1.5 to 3.0 kb. In this study, the centromere from chromosome 5 was used. Splitting of the chromosome was carried out by two steps as follows: specific cleavage of the chromosome by the Cas9-sgRNA complex and integration of the telomere and centromere at the cleaved ends of chromosome by HDR (Fig 1). When the chromosome was cleaved, centric and acentric chromosome fragments were produced. The ends of both those centric and acentric chromosomes would be protected by the introduced telomere. The centric chromosome would segregate accurately into daughter cells, but the acentric chromosome would not, due to the lack of centromere (Fig 1). Thus, to make acentric chromosome segregate, the addition of centromere would be necessary.

We selected PBANKA_0111600, which encodes a rhoptry protein (ROP14) as a target for the cleavage site (Fig 2A). The ROP14 which is located at 444,999–448,675 of chromosome 1, and since it is not essential for asexual development in RBC, it can be disrupted without deleterious effects in this developmental stage [15]. The sgRNA plasmid specific for *rop14* and the two linear donor templates were co-introduced into the pbcas9 parasite, in which *cas9* of *Streptococcus pyogenes* was integrated at the *cssu* locus (Fig 2A) [10]. Parasites emerged in peripheral blood 2 days after removal of drug selective pressure, and subsequent genotyping analysis by PCR indicated the presence of the transgenic parasites with split chromosomes, while there was also the wild-type parasites present. The parasite line with split chromosomes were further cloned by limiting dilution and named as split-Ch1-1116. The genotyping analysis of split-Ch1-1116 parasite indicated the splitting of chr1 (Fig 2B).

To confirm the split of chromosome 1, we performed CHEF-electrophoresis of split-Ch1-1116, followed by Southern hybridization using probes specific to each split chromosome fragment: two DNA probes were derived from PBANKA_0104900 and PBANKA_0112500, which are located on the left and the right chromosome fragments, respectively (Figs 2C and S2A). The signals of both left and right chromosome fragments were detected at the expected sizes of

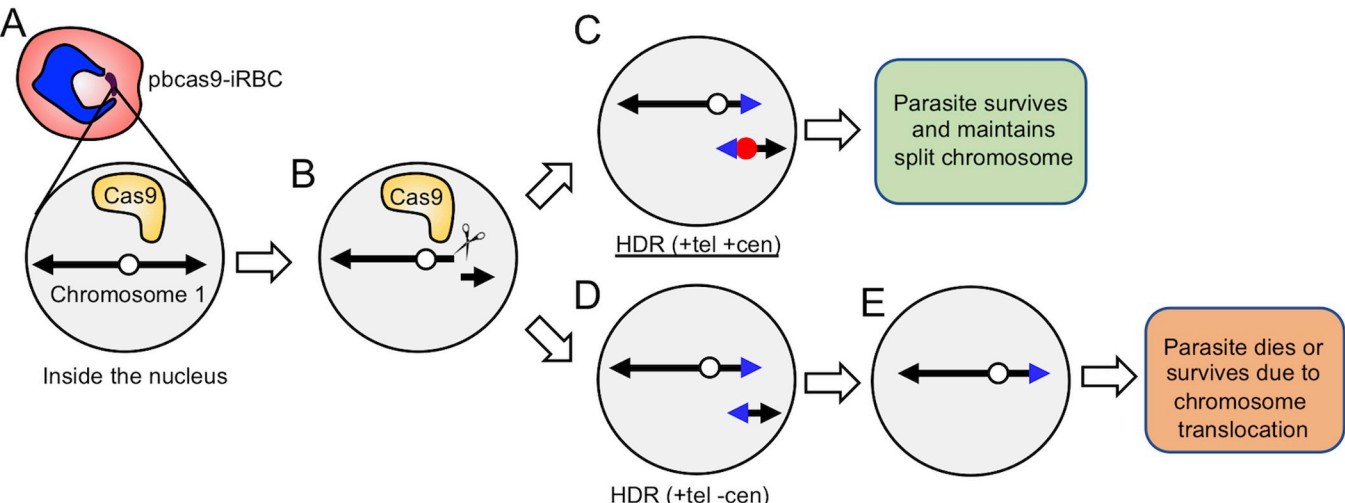

**Fig 1. Experimental scheme of chromosome splitting.** (A) pbcas9 is the transgenic parasite expressing the Cas9 nuclease in the nucleus. (B) Chromosome 1 is cleaved by the Cas9, which generates a centric fragment and an acentric fragment. The white circle is the original centromere. (C) The cleaved end of the centric chromosome fragment is repaired by homologous recombination using the donor DNA with a telomere sequence (blue triangle). While, that of the acentric chromosome fragment is repaired using the donor template with both a centromere (red circle) and telomere (blue triangle). The split chromosomes will be maintained in the parasite due to the centromere and telomere. (D and E) In contrast, if the cleaved end of the acentric chromosome fragment is repaired using the donor template with only a telomere, the parasite will lose the split chromosome due to the failure of its segregation. As a result, the parasite will die due to this loss of a chromosome.

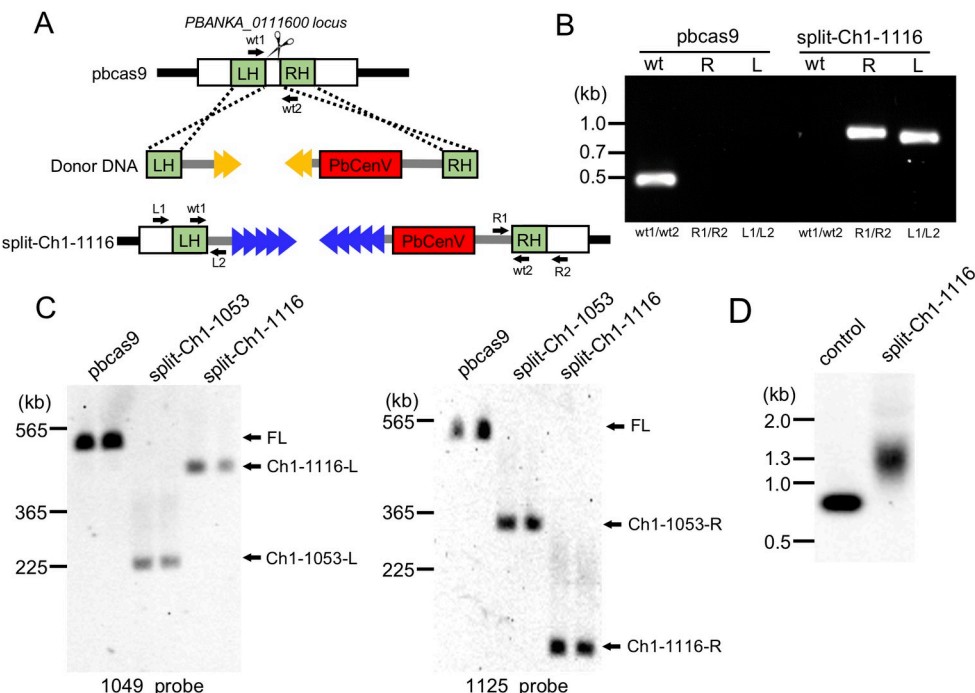

**Fig 2. CRISPR/Cas9-based chromosome split.** (A) The PBANKA_011600 was cleaved by the Cas9-sgRNA complex, followed by HDR with the donor template DNA including only telomere, and telomere and centromere. (B) The genotyping PCR was performed using the sets of primers indicated at the bottom. (C) Southern hybridization analyses of the transgenic parasites, which are split-Ch1-1116 and -1053, detected the split chromosomes. The probe DNA used in each analysis is shown at the bottom. The information about the probe are described in S2 and S3 Figs. (D) Southern hybridization shows the telomere extension in the transgenic parasite.

approximately 450 kbp and 60 kbp, respectively (Fig 2C). Similar results were obtained from another biologically independent split-Ch1-1116 parasite. These results demonstrate that chromosome could be divided by CRISPR/Cas9 system using telomere and centromere sequences.

In the previous study, when a telomere sequence was added to the end of cleaved chromosome, *de novo* elongation of the telomere occurred [10]. To investigate whether a similar elongation would be observed in this study, we performed Southern hybridization analysis of the telomere end of the split chromosome. The signal derived from the right chromosome fragment was detected at approximately 1.3 kbp, when the digested genomic DNA of split-Ch1-1116 was hybridized with the probe DNA from the proximal region of the additional telomere (Figs 2D and S2B). On the other hand, in the negative control using the linear donor template, a signal was detected at 0.8 kbp (Figs 2D and S2B). This size difference between those two signals indicated the *de novo* elongation of the telomere at the end of split chromosome. In addition, the signal is broad, and its width is maintained within about 0.4 kbp, suggesting that *de novo* telomere extension is regulated within a certain length. These results suggested that the introduced telomere sequence was recognized by telomerase in split-Ch1-1116 like the original telomere, allowing for the stable maintenance of the split chromosome in the parasites.

To examine the versatility of this method, we split chromosome 1 at another genomic locus, where PBANKA_0105300 was located (S3A Fig). The PBANKA_0105300 is located at 216,483–221,962 of chromosome 1, encodes the unknown-function *Plasmodium* protein and is not essential for asexual development in RBCs [15]. Transfection experiments using the plasmid having the sgRNA and two linear donor templates were performed, and the clonal

transgenic parasite obtained by the limiting dilution was named split-Ch1-1053 (S3B Fig). The genotyping and Southern hybridization analyses of split-Ch1-1053 demonstrated the splitting of chromosome 1: PCR products specific for the split left and right chromosome fragments were amplified (S3C Fig). In addition, the signals derived from those two fragments were detected at expected sizes, which were approximately 365 kbp and 225 kbp (Fig 2C). Furthermore, the elongation of the *de novo* telomere was confirmed by Southern analysis (S3D and S3E Fig). These results showed that any genomic locus of interest could be split using this method.

## Effect of splitting the chromosome on the development of parasites

*Plasmodium* parasites undergo atypical mitotic division during asexual multiplication in RBCs, whereby a multinucleate syncytium, schizont, is formed. All 14 chromosomes segregate accurately into those divided nuclei in schizonts, followed by the formation of RBC-invasive merozoite. As a result of splitting the chromosome, the total number of chromosomes became 15, which might have some influence on mitotic division. To examine this, we compared the asexual multiplication of split-Ch1-1116 parasite to that of pbcas9. The result clearly showed comparable growth of both parasite lines in RBCs. The growth rates of split-Ch1-1116 and pbcas9 were estimated as 5.6 and 6.4, respectively, indicating that there was no significant defect of asexual multiplication in split-Ch1-1116 (Fig 3A). Moreover, split-Ch1-1116 formed a comparable number of nuclei in schizonts alongside pbcas9, and no defects in their ring forms and trophozoites other than schizonts were observed (Fig 3B and 3C). Taken together,

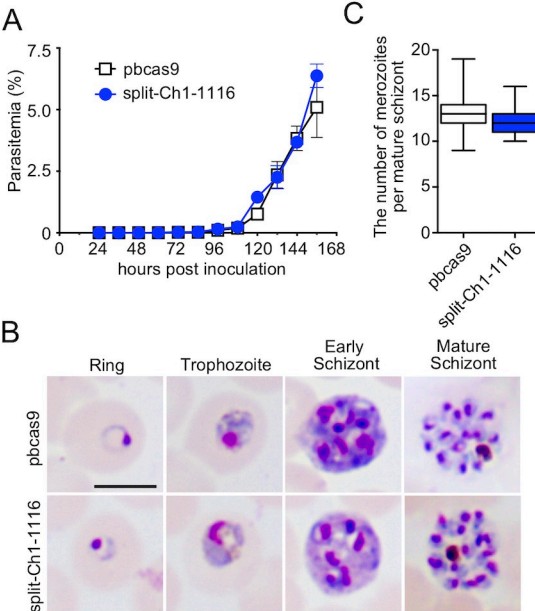

**Fig 3. Asexual development of the split-Ch1-1116 parasites.** (A) The growth of the split-Ch1-1116 parasites, which is indicated by the blue line, was comparable to that of the parental pbcas9 parasites, which is shown by the black line. The points and error bars represent the mean and standard error of the mean of triplicate values. Distributions for each day were compared using the unpaired *t*-test (not significant). (B) The morphologies of the parasites during asexual development were similar between pbcas9 (upper) and split-Ch1-1116 (lower). The bar indicates 5 μm. (C) The number of merozoites per schizonts of the split-Ch1-1116 parasite was comparable to that of the parental pbcas9 strain. The middle line, top, and bottom of the box, top and bottom whiskers are the median, 75th and 25th percentiles, and the maximum and minimum values respectively. Distributions were compared using the unpaired *t*-test (not significant).

these results showed that an increase in chromosome number due to splitting did not affect mitotic division of the parasites, and that the parasite was able to multiply asexually in RBCs without any defects, despite its chromosome being split.

We next examined whether splitting the chromosome affects sexual development and fertilization. It is known that some part of the parasites undergoes sexual commitment during asexual development in RBCs, then develop into female and male gametocytes. These gametocytes fertilize in the midgut of *Anopheles* mosquitos after blood feeding and develop into zygotes, followed by the formation of ookinetes which is a midgut invasive form. Meiosis occurs in zygotes, and the parasites then return to haploid from diploid [16]. Here, we observed both female and male gametocytes of split-Ch1-1116 in peripheral blood. In addition, exflagellation of male gametes was induced by xanthurenic acid, and the number of ex-flagellation centres of split-Ch1-1116 were comparable to that of pbcas9 (Fig 4A). A comparison of the morphologies of the ookinetes in giemsa-stain smears for both split-Ch1-1116 and pbcas9 parasites showed no significant difference (Fig 4B), suggesting not only normal ookinete formation, but also zygote development. Furthermore, the average of mature ookinete rates was comparable in split-Ch1-1116 (86.4%) to the pbcas9 parasites (85.0%) (Fig 4C). These results suggested that splitting the chromosome did not interfere with asexual multiplication, sexual development, fertilization, and ookinete formation including meiosis.

## Discussion

Elucidating the mechanism of the regulation of gene expression is essential for understanding parasite's complex life cycle. Recent advanced analysis, such as Hi-C analysis suggests that, in addition to sequence-specific transcriptional factors and epigenetic regulators, the spatial arrangement of chromosomes is involved in the regulation of gene expression [4, 5]. It is possible that this spatial arrangement is altered by chromosome splitting using our method, which may cause a gene expression and epigenetic state change: since telomere ends are invariably anchored to inner nuclear membrane [17], the *de novo* telomere generated as a result of

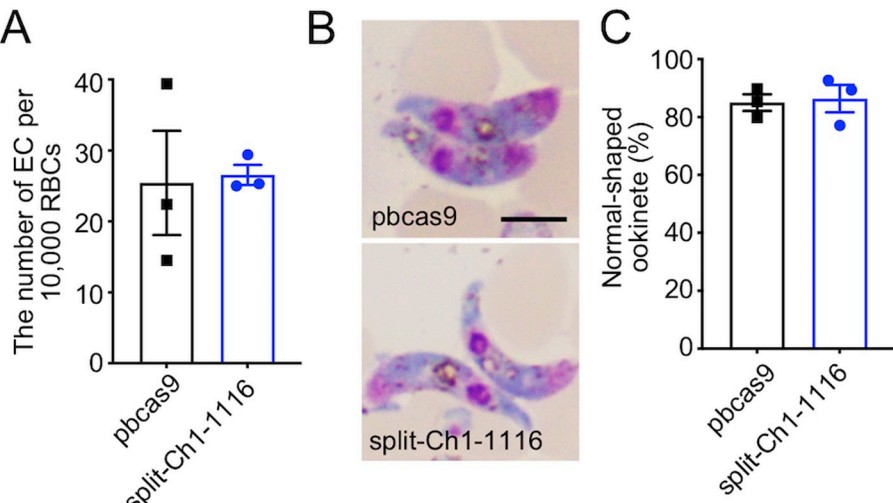

**Fig 4. Sexual development of the split-Ch1-1116 parasites.** (A) The number of exflagellation centers of split-Ch1-1116 parasites were comparable to that of pbcas9 parasites. The column and error bars indicate the mean and standard error from biological triplicates. (B and C) Ookinete shape and conversion rate were normal in the split-Ch1-1116 parasites. The columns and error bars indicate the mean and standard error of the mean from biological triplicates. Distributions in A and C were compared using the unpaired *t*-test (not significant). The bar indicates 5 μm.

splitting chromosome may be also tethered to inner nuclear membrane, and this tethering of the *de novo* telomere may cause large-scale rearrangement of chromosomes. As our method allows the generation of *de novo* telomere ends at any genomic loci, it is possible to alter spatial arrangement of chromosomes specifically at locus of interest. For example, the spatial arrangement of heterochromatic regions, where infected RBC-surface antigens and the sex-specific transcription factor *ap2-g* are located, may be altered by our method, which may allow us for investigating mechanisms of host immune evasion and sexual development of parasites. Therefore, our method will assist for further understanding of parasite's life cycle from the view of spatial arrangement of chromosomes.

The normal mitotic division of split-Ch1-1116 parasites in RBCs indicated that *de novo* centromere functioned properly: it acted as the specific genomic site for assembling kinetochores which is the protein complex to direct chromosome movement along spindle microtubule [18]. The formation of kinetochore requires a centromere-specific histone H3, called CENP-A [19]. Thus, the CENP-A must be loaded specifically on the *de novo* centromere of the split chromosome and maintained epigenetically. This specific loading of CEMP-A highly depends on the sequence of centromere. Since the linear donor template containing centromere was nucleosome-free DNA fragment, there is no factors other than the sequence. This unique sequence dependency of CENP-A loading of *Plasmodium* parasite is considered to be one of the factors for successful splitting of the chromosome. Our Southern blotting analyses of split-Ch1-1116 and -1053 showed that *de novo* telomeres were elongated by telomerase and their length was regulated constantly. These results indicated that *de novo* telomeres were recognized by not only telomerase, but also other telomere binding proteins [20, 21], similar to the original telomere. This recognition by telomere-associated molecules allowed the introduced telomeric sequences to function normally, protecting the ends of the split chromosomes and contributing to their stability in transgenic parasites. Therefore, we consider that this could be another factor for splitting chromosomes.

In conclusion, we demonstrated that the chromosome could be split by CRISPR/Cas9 system using telomere and centromere. In contrast to this method, there is a method for the fusion of chromosome ends using CRISPR/Cas9 system: all chromosomes were fused in *Saccharomyces cerevisiae*, which generated one or two chromosomes encoding all genes. Theoretically, a similar method could be developed in *P. berghei*. Combining these splitting/fusing methods, the chromosomes will be rearranged in large-scale which will provide us a strong tool for investigating the role of the spatial arrangement of chromosomes in gene expression.

## Supporting information

**S1 Table. Oligonucleotides used in this study.**
(XLSX)

**S1 Fig. Maps of donor plasmids constructed for the chromosome split experiment.**
(TIFF)

**S2 Fig. Illustrations of split chromosome 1 and de novo telomere end of split site.**
(TIFF)

**S3 Fig. CRISPR/Cas9-based chromosome split in the split-Ch1-1053 parasites.**
(TIFF)

**S1 File. Original gel and blot images.**
(PDF)

## Acknowledgments

We thank Mr. Takashi Sekine and Dr. Rie Kubota for technical support with Southern hybridization analysis. We are also grateful to Dr. Ryoichi Saito for equipment support with CHEF electrophoresis analysis.

## Author Contributions

**Conceptualization:** Shiroh Iwanaga.

**Investigation:** Daniel Addo-Gyan, Haruka Matsushita, Enya Sora, Tsubasa Nishi, Naoaki Shinzawa.

**Writing – original draft:** Masao Yuda.

**Writing – review & editing:** Daniel Addo-Gyan.

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
