## [Decision Letter · Decision Letter 0]

13 Dec 2021

PONE-D-21-34882Chromosome splitting of Plasmodium berghei using the CRISPR/Cas9 system.PLOS ONE

Dear Dr. Shiroh Iwanaga,

Thank you for submitting your manuscript to PLOS ONE. After careful consideration, we feel that it has merit but does not fully meet PLOS ONE’s publication criteria as it currently stands. Therefore, we invite you to submit a revised version of the manuscript that addresses the points raised during the review process. First, I would like to apologize for the time taken to arrive at a decision (Minor Revision) that was due to any potential qualified reviewers declining our invitation to review your submission. One person did accept and luckily she is a real expert in P. berghei chromosome modifications and so, I took the editorial decision based on only this single review. When submitting your revised manuscript please make absolutely clear in your rebuttal letter how you addressed her comments, as this will allow me to make a rapid editorial decision without sending your revision back out for review.

We look forward to receiving your revised manuscript.

Kind regards,

Gordon Langsley

Academic Editor

PLOS ONE

Journal Requirements:

2. To comply with PLOS ONE submissions requirements, in your Methods section, please provide additional information on the animal research and ensure you have included details on (1) methods of sacrifice, (2) methods of anesthesia and/or analgesia, and (3) efforts to alleviate suffering.

4. We note that Figures 3 and 4 in your submission contain copyrighted images. All PLOS content is published under the Creative Commons Attribution License (CC BY 4.0), which means that the manuscript, images, and Supporting Information files will be freely available online, and any third party is permitted to access, download, copy, distribute, and use these materials in any way, even commercially, with proper attribution. For more information, see our copyright guidelines: http://journals.plos.org/plosone/s/licenses-and-copyright.

a. You may seek permission from the original copyright holder of Figures 3 and 4 to publish the content specifically under the CC BY 4.0 license. 

All studies were supported by Grants-in-Aid for Scientific Research (20H03477 and 19K22527 to S.I. 18K07084 and 21K06985 to N.S., and 17H01542 to M.Y.), which were funded by the Japan Society for the Promotion of Science (JSPS: https://www.jsps.go.jp/), and also supported by the Japan Agency for Medical Research and Development (AMED: https://www.amed.go.jp/) under Grant Number 21jm0210061h0004 (to S.I.), 21wm0325018 (to N.S.) and 21wm0225014 (to N.S.).

Reviewers' comments:

Reviewer's Responses to Questions

**Comments to the Author**

1. Is the manuscript technically sound, and do the data support the conclusions?

Reviewer #1: Yes

2. Has the statistical analysis been performed appropriately and rigorously? 

Reviewer #1: Yes

3. Have the authors made all data underlying the findings in their manuscript fully available?

Reviewer #1: Yes

4. Is the manuscript presented in an intelligible fashion and written in standard English?

Reviewer #1: Yes

5. Review Comments to the Author

Reviewer #1: Summary

The authors here present a study where they have generated a CRISPR/Cas9-based tool which facilitates the site-specific cleavage of Plasmoidum berghei chromosomes. By introduction of additional telomere and centromere elements together with the homology directed repair template, they can thereby generate transgenic parasite lines with split chromosomes that segregate and are faithfully maintained during cell division. The study builds upon improvements to the P. berghei CRISPR/Cas9 system that the some of the authors published earlier. They here report on the on the modifications to the system that enables chromosome splitting, the vector generation and proof of principle experiments where they split P. berghei chromosome 1 in two different (blood-stage) functionally redundant loci. They further report on the normal viability and lack of fitness effect for (one of the?) split chromosome 1 lines.

Impact and quality

This is a well written manuscript with clearly presented methods and results, and where conclusions are well supported by data. Data is of good quality and is underpinned by appropriate controls. The merit of this study is that it presents a for malaria parasites completely novel system to split chromosomes and generate viable progeny with split chromosomes. This tool in turn, as argued by the authors, makes it possible to study the effect of intra-nuclear spatial location of chromosomes, which the authors argue “may cause a gene expression and epigenetic state change: since telomere ends are invariably anchored to inner nuclear membrane”. They further suggest this will be a particularly useful tool to study the effect of intra-nuclear spatial localization on expression of multigene surface proteins and genes important for sexual development since they localize to the subtelomeric regions of malaria parasite chromosomes.

The impact of the study is limited by that the authors present no evidence for change in intra-nuclear spatial location of split chromosomes (e.g. FISH) or of gene expression (e.g. RNA seq) due to chromosome splitting. Could one not instead argue that no change in gene expression is likely to occur due to chromosome splitting, since no observable change in phenotype is shown for the chromosome 1 split line? The impact of this system on our understanding of malaria parasite biology there for remain hypothetical. This type of data would significantly elevate the impact of the study. Nevertheless, that elevation would likely qualify it for a higher impact journal and I have no reservations in recommending the manuscript for publication as it is, with only minor revisions outlined below.

Minor revisions

41 considered recently -> recently considered

95 this plasmid can be screened -> this plasmid can be selected?

103 perhaps will aid reader if you explain centric / acentric chromosome?

277 “… we compared the asexual multiplication of split-Ch1-S parasite to that of pbcas9”. Clarify here if split-Ch1-S is PBANKA_0111600 or PBANKA_0105300 line. Perhaps it is stipulated in methods or elsewhere but would be good to make more obvious. This goes for all figures too, it is not always immediately clear to the reader which split chromosome line you refer to.

Fig S4 Mislabeling, two Panel D instead of E

6. PLOS authors have the option to publish the peer review history of their article (what does this mean?). If published, this will include your full peer review and any attached files.

Reviewer #1: **Yes: **Ellen Bushell

---

## [Author Response · Author response to Decision Letter 0]

18 Jan 2022

Reviewer’s Comments

Reviewer #1: Summary

The authors here present a study where they have generated a CRISPR/Cas9-based tool which facilitates the site-specific cleavage of Plasmoidum berghei chromosomes. By introduction of additional telomere and centromere elements together with the homology directed repair template, they can thereby generate transgenic parasite lines with split chromosomes that segregate and are faithfully maintained during cell division. The study builds upon improvements to the P. berghei CRISPR/Cas9 system that the some of the authors published earlier. They here report on the on the modifications to the system that enables chromosome splitting, the vector generation and proof of principle experiments where they split P. berghei chromosome 1 in two different (blood-stage) functionally redundant loci. They further report on the normal viability and lack of fitness effect for (one of the?) split chromosome 1 lines.

Impact and quality This is a well written manuscript with clearly presented methods and results, and where conclusions are well supported by data. Data is of good quality and is underpinned by appropriate controls. The merit of this study is that it presents a for malaria parasites completely novel system to split chromosomes and generate viable progeny with split chromosomes. This tool in turn, as argued by the authors, makes it possible to study the effect of intra-nuclear spatial location of chromosomes, which the authors argue “may cause a gene expression and epigenetic state change: since telomere ends are invariably anchored to inner nuclear membrane”. They further suggest this will be a particularly useful tool to study the effect of intra-nuclear spatial localization on expression of multigene surface proteins and genes important for sexual development since they localize to the subtelomeric regions of malaria parasite chromosomes.  The impact of the study is limited by that the authors present no evidence for change in intra-nuclear spatial location of split chromosomes (e.g. FISH) or of gene expression (e.g. RNA seq) due to chromosome splitting. Could one not instead argue that no change in gene expression is likely to occur due to chromosome splitting, since no observable change in phenotype is shown for the chromosome 1 split line? The impact of this system on our understanding of malaria parasite biology there for remain hypothetical. This type of data would significantly elevate the impact of the study. Nevertheless, that elevation would likely qualify it for a higher impact journal and I have no reservations in recommending the manuscript for publication as it is, with only minor revisions outlined below.

Response:

Thank you for reviewing our manuscript to make it better. The details of the modifications according to your comments are described below.

Minor revisions

1. 41 considered recently -> recently considered

Response

Thank you for pointing-out this grammatical error. We have modified the phase from “considered recently” to “recently considered” on line 41.

2. 95 this plasmid can be screened -> this plasmid can be selected?

Response

We have changed the word “screened” to “selected” on line 98.

3. 103 perhaps will aid reader if you explain centric / acentric chromosome?

Response

Thank you for this suggestion. We have included an explanation in parenthesis for both acentric and centric chromosomes. This can be seen from lines 108 to 112 as follows:

This linear DNA fragment was used for HDR with the centric chromosome (Chromosome 1 fragment with the original centromere after cleavage) after cleavage by the Cas9-sgRNA complex. Another DNA fragment, which was used for HDR with the acentric chromosome (Chromosome 1 fragment lacking a centromere after cleavage), was amplified and cloned upstream of both telomere and centromere of pArm_R plasmid (S2 Fig)” (P.5,l.106-P.6,l.110)

4. 277 “… we compared the asexual multiplication of split-Ch1-S parasite to that of pbcas9”. Clarify here if split-Ch1-S is PBANKA_0111600 or PBANKA_0105300 line. Perhaps it is stipulated in methods or elsewhere but would be good to make more obvious. This goes for all figures too, it is not always immediately clear to the reader which split chromosome line you refer to.

Response

Thank you for this observation. In order to make it easier to understand which mutant parasite is being referred-to, the labeling for the mutants have been modified. For example, split-Ch1-S has been changed to split-Ch1-1116. “1116” indicates that the mutant was produced by splitting the parent parasite at the PBANKA_0111600 locus. Similarly, split-Ch1-L has been changed to split-Ch1-1053. Also, “1053” indicates that the mutant was produced by splitting the parent parasite at the PBANKA_0105300 locus. This change has been effected throughout the text and figures.

5. Fig S4 Mislabeling, two Panel D instead of E

Response

Thank you for pointing-out this mistake. The mislabeled “D” panel has been re-labeled as “E” in the S4 Fig caption.

---

## [Editor Report · Decision Letter 1]

9 Feb 2022

Chromosome splitting of Plasmodium berghei using the CRISPR/Cas9 system.

PONE-D-21-34882R1

Dear Dr. Shiroh Iwanaga,

We’re pleased to inform you that your manuscript has been judged scientifically suitable for publication and will be formally accepted for publication once it meets all outstanding technical requirements.

Kind regards,

Gordon Langsley

Academic Editor

PLOS ONE
---

## [Editor Report · Acceptance letter]

14 Feb 2022

PONE-D-21-34882R1 

Chromosome splitting of *Plasmodium berghei* using the CRISPR/Cas9 system. 

Dear Dr. Iwanaga:

I'm pleased to inform you that your manuscript has been deemed suitable for publication in PLOS ONE. Congratulations! Your manuscript is now with our production department. 

Kind regards, 

on behalf of

Dr. Gordon Langsley 

Academic Editor

PLOS ONE